# Comparative Multi-Omics Analysis Reveals Lignin Accumulation Affects Peanut Pod Size

**DOI:** 10.3390/ijms232113533

**Published:** 2022-11-04

**Authors:** Zhenghao Lv, Dongying Zhou, Xiaolong Shi, Jingyao Ren, He Zhang, Chao Zhong, Shuli Kang, Xinhua Zhao, Haiqiu Yu, Chuantang Wang

**Affiliations:** 1Peanut Research Institute, College of Agronomy, Shenyang Agricultural University, Shenyang 110000, China; 2Shandong Peanut Research Institute, Shandong Academy of Agricultural Sciences, Qingdao 266000, China

**Keywords:** peanut (*Arachis hypogaea* L.), pod size, metabolome, transcriptome, lignin biosynthesis

## Abstract

Pod size is one of the important factors affecting peanut yield. However, the metabolites relating to pod size and their biosynthesis regulatory mechanisms are still unclear. In the present study, two peanut varieties (Tif and Lps) with contrasting pod sizes were used for a comparative metabolome and transcriptome analysis. Developing peanut pods were sampled at 10, 20 and 30 days after pegging (DAP). A total of 720 metabolites were detected, most of which were lipids (20.3%), followed by phenolic acids (17.8%). There were 43, 64 and 99 metabolites identified as differentially accumulated metabolites (DAMs) at 10, 20 and 30 DAP, respectively, and flavonoids were the major DAMs between Tif and Lps at all three growth stages. Multi-omics analysis revealed that DAMs and DEGs (differentially expressed genes) were significantly enriched in the phenylpropanoid biosynthesis (ko00940) pathway, the main pathway of lignin biosynthesis, in each comparison group. The comparisons of the metabolites in the phenylpropanoid biosynthesis pathway accumulating in Tif and Lps at different growth stages revealed that the accumulation of p-coumaryl alcohol (H-monolignol) in Tif was significantly greater than that in Lps at 30 DAP. The differential expression of *gene-LOC112771695*, which is highly correlated with p-coumaryl alcohol and involved in the biosynthesis of monolignols, between Tif and Lps might explain the differential accumulation of p-coumaryl alcohol. The content of H-lignin in genetically diverse peanut varieties demonstrated that H-lignin content affected peanut pod size. Our findings would provide insights into the metabolic factors influencing peanut pod size and guidance for the genetic improvement of the peanut.

## 1. Introduction

Peanut (*Arachis hypogaea* L.) is one of the most important oilseed and cash crops cultivated worldwide. In recent years, the global demand for vegetable oils has markedly increased. Meanwhile, as the main source of vegetable oil in China, peanut production is still waiting to be improved [1]. Therefore, increasing yield is one of the most important topics of peanut breeding programs. Yield is a complex quantitative trait, and for peanuts, pod size directly determines the final yield. The peanut pod is composed of shell and seed, and the main components of peanut shell include holocellulose, lignin, ash and organic solvent extracts (OSE) [2]. Compared with the seed, the peanut shell develops first and acts as a protective and perceived organ to ensure the normal development of the seeds [3]. On the other hand, the peanut endocarp is a temporary place, used for storing photosynthetic products for seed development; the transport of photosynthate to seed determines seed weight and filling [4]. Consequently, the peanut shell affects seed development, and pod size is directly determined by the swelling of the shell.

Different from other legumes, peanut has the characteristic of “aerial flower, subterranean fruit”. After fertilization, peanut zygotes form a pre-embryo after several divisions, then stop growing, and the intercalary meristem begins to divide to form a peg (gynophore) [5,6]. The embryo development is resumed and the pod is formed after peg penetration into the soil [7,8]. Subsequently, the tip region of the peg develops into a globular-stage embryo, the peg stops elongating, and the pod begins to expand [9]. Previous studies have shown that the final plant fruit size is determined early during the growth stage [10,11]. For peanut, pod-expansion generally begins at about 10 days after pegging (DAP), then grows rapidly from 10 to 20 DAP, and finally reaches its final size between 20 and 30 DAP. During this period, darkness, mechanical stimulation, moisture and nutrition promoted the development of pod. This process is the crucial determinant of peanut yield [12].

Lignin is a complex heterogeneous polymer derived from three monolignols (p-coumaryl, coniferyl, and sinapyl alcohol). The composition and structure of lignin varies greatly within and among plants [13]. The deposition mode of para-hydroxy-phenyl lignin (H-type lignin) is different from that of guaiacyl lignin (G-type lignin) or syringyl lignin (S-type lignin), which suggests H-type lignin has special effects. H-type lignin is thought to be located in the outermost layer of the cell wall and determines the cell shape [14]. Monolignols are synthesized by the phenylpropanoid pathway and then oxidatively polymerized to lignin polymers [15]. The downregulation of the expression of monolignol biosynthesis-related genes leads to the reduction of lignin content [16]. For instance, the inhibition of *Os4CL3* gene expression significantly reduced lignin content in rice [17]. In maize, transgenic plants with downregulated *COMT* expression showed a strong decrease in Klason lignin content and lower p-coumaric acid content [18]. Moreover, several studies revealed that there is a negative relationship between the lignin content and the growth and development of plants [19,20]. In peanut, the lignification process is speculated to affect the width of pod [21].

With the advance of sequencing technology and rapid reduction in costs, transcriptome sequencing (RNA-Seq) has been widely used in gene expression measurement [22]. Plant metabolites reflect their responses to biological and abiotic stimuli, serving as a connecting link between genotype and phenotype [23]. The combined analysis of transcriptome and metabolomics can yield vital information on the complex biological regulation of target traits. However, studies of the metabolomics of the peanut pod have rarely been reported. In this study, a widely targeted metabolomics profiling of Tif and Lps pods was performed at different growth stages (10, 20 and 30 DAP) to study the accumulation of metabolites and to identify metabolites that could contribute to pod development. At the same time, the transcriptomes of each growth stage were analyzed to study the genetic control of the differential accumulation of these metabolites. Overall, our study will contribute to the understanding of the mechanisms in peanut pod development, reveal the genes and metabolites that are responsible for determining pod size and provide a foundation for the genetic improvement of peanut pod.

## 2. Results

### 2.1. Phenotypic Differences in Pod Size between Tif and Lps

Peanut pod expansion is generally performed at the early growth stage (10–30 DAP), during which the pod reaches its final size. In this study, the pod length and width of Tif and Lps were measured at 10, 20 and 30 DAP. The results showed that the pod length and width of Lps were significantly greater than those of Tif at the three growth stages (Figure 1A). Among these, the pod length of Lps was 25.8, 14.6 and 27.6% longer than that of Tif at 10, 20 and 30 DAP, respectively (Figure 1B). For pod width, Lps was 34.0, 11.4 and 23.9% wider than that of Tif at each growth stage, respectively (Figure 1C). Furthermore, the pod length and width of both Tif and Lps increased rapidly from 10 to 20 DAP. During the 20 to 30 DAP period, the pod length and width of Lps still showed an obvious growth trend of 27.2 and 28.5%, respectively. However, this growth trend was not obvious in Tif, and the pod length and width increased by 14.2 and 15.4% from 20 to 30 DAP, respectively.

### 2.2. Metabolomic Analysis of Peanut Pods

A widely targeted metabolomics analysis was used to evaluate the differences in metabolites between Tif and Lps during pod development. In total, 720 metabolites were detected (Appendix A), including 146 lipids (20.3%), 128 phenolic acids (17.8%), 77 amino acids and derivatives (10.7%), 66 organic acids (9.2%), 59 nucleotides and derivatives (8.2%), 50 alkaloids (6.9%), 39 terpenoids (5.4%), 38 flavonoids (5.3%), 25 lignans and coumarins (3.5%), 4 quinones (0.6%) and 88 other metabolites (12.2%) (Figure 2A). The principal component analysis (PCA) of all the identified metabolites showed that the first two principal components explained about 74% of the total variation (PC1 = 50.26%, PC2 = 23.73%) (Figure 2B). The similarity between the three replicates of each group was high, indicating that the analysis results were stable and repeatable. In the PCA scatter plot, the pod samples with the same growth stage were clustered well, but not clustered according to different materials. This phenomenon indicated that significant changes had taken place in the metabolites at the different growth stages of the peanut pods. The heatmap of all identified metabolites is shown in Figure 2C. All the samples were hierarchically clustered into three main branches (10, 20 and 30 DAP), indicating that the accumulation of metabolites varied at different growth stages, which was consistent with the PCA analysis. Moreover, most of the metabolites displayed early accumulation in both Tif and Lps, and the compounds at 10 DAP mainly comprised lipids, phenolic acids, amino acids and derivatives, and nucleotides and derivatives, reflecting the biochemical accumulation in the peanut pods in the early stage. After that, the accumulation of metabolites decreased as development proceeded.

### 2.3. Identification of Differentially Accumulated Metabolites between Tif and Lps

In the OPLS-DA models (Appendix A), Tif10DAP, Tif20DAP and Tif30DAP clearly separated from Lps10DAP, Lps20DAP and Lps30DAP, respectively, suggesting major distinctions in the metabolic profiles between Tif and Lps. We further performed differentially accumulated metabolite (DAM) screening of all metabolites at the three growth stages based on the log2fold-change (|log2FC| > 1) variables identified as important in the projection (VIP > 1) scores and *p*-value < 0.05. There were 43 (18 upregulated and 25 downregulated), 64 (37 upregulated and 27 downregulated) and 99 (41 upregulated and 58 downregulated) metabolites identified as DAMs between Tif and Lps at 10, 20 and 30 DAP, respectively (Figure 3A–C). This trend indicated that the number of DAMs increased as pod development proceeded. Meanwhile, 14 flavonoids (27.5%), 12 phenolic acids (23.5%) and 7 alkaloids (13.7%) were major DAMs at 10 DAP (Tif10DAP vs. Lps10DAP); 15 flavonoids (22.4%), 15 phenolic acids (22.4%) and 9 terpenoids (13.4%) were major DAMs at 20 DAP (Tif20DAP vs. Lps20DAP); 15 flavonoids (14.2%), 27 phenolic acids (25.5%) and 13 alkaloids (12.3%) were major DAMs at 30 DAP (Tif30DAP vs. Lps30DAP) (Appendix A). In addition, the k-means clustering analysis divided the DAMs into eight distinct clusters (G1–G8) (Figure 3E), suggesting that the accumulation patterns of the DAMs were diverse throughout pod development. These clusters could be classified into two categories. The metabolites with the same accumulation trend in Tif and Lps (G3, G4, G6 and G8) were classified into one category. Four clusters in this category corresponded to three different growth stages: 10 DAP (G3), 20 DAP (G4 and G6), and 30 DAP (G8). The metabolites with different accumulation trends in Tif and Lps (G1, G2, G5 and G7) were classified into another category. From inspection of the functions of clustered metabolites, we found that clustered metabolites were involved in a variety of metabolism and biosynthesis pathways (Figure 3F). These analyses revealed that the metabolism pathways of purine, nucleotide, thiamine, and tryptophan as well as the biosynthesis of phenylpropanoid, flavone, flavonol, and flavonoid were mainly carried out at 10 DAP; the metabolism pathways of purine, carbon, phosphonate and phosphinate as well as the biosynthesis of phenylpropanoid, flavonoid, stilbenoid, diarylheptanoid and gingerol, and the pentose phosphate pathway were mainly carried out at 20 DAP; the metabolism pathways of tryptophan, sulfur, tyrosine, arginine and proline as well as the biosynthesis of benzoxazinoid phenylpropanoid, flavone, and flavonol were mainly carried out at 30 DAP. These major metabolic and biosynthesis pathways determined changes in biochemical compounds during peanut pod development.

### 2.4. Transcriptome Analysis Revealed Multiple Biological Processes Involved in Pod Development

A RNA-Seq analysis was performed during the process of pod development, including 10, 20 and 30 DAP. |log2FC| ≥ 1 and FDR value < 0.05 were used as threshold to identify differentially expressed genes (DEGs). There were 5628 DEGs (3061 upregulated and 2567 downregulated), 7378 DEGs (3778 upregulated and 3600 downregulated), and 4346 DEGs (2218 upregulated and 2128 downregulated) identified at 10, 20, and 30 DAP, respectively (Appendix A). Similar to clusters from the metabolic analysis, the k-means clustering of the DEGs exhibited eight distinct clusters (T1–T8) (Figure 4A). While inspecting the potential function of DEGs Figure 4B), the high-expression genes in both Tif and Lps at 10DAP (T1 and T8) were mainly involved inthe cell wall, anchored component of membrane and plasma membrane, DNA replication, which may be related to the metabolic and biosynthesis processes involving flavonoid, phenylpropanoid and lignin enriched in the G3 cluster. The high-expression genes at 20 DAP (T5) and 30 DAP (T6 and T7) were mainly involved in the metabolic processes of the cell wall macromolecule, cell wall polysaccharide, and phenylpropanoid as well as the biosynthesis of cell wall and secondary cell wall. Compared with metabolomic and transcriptomic data, our results showed that the differential expression of identified genes was highly associated with changes in metabolic accumulation with pod development.

### 2.5. A Conjoint Analysis of Transcriptome and Metabolome

In order to investigate the relationships between genes and metabolites involved in the process of pod development, the DAMs and DEGs in the three comparison groups (Tif10 DAP vs. Lps10 DAP, Tif20 DAP vs. Lps20 DAP and Tif30 DAP vs. Lps30 DAP) were mapped according to the KEGG database. There were 14, 19 and 33 co-mapped pathways in the three comparison groups, respectively (Figure 5A–C). The results showed that genes or metabolites were significantly enriched in the phenylpropanoid biosynthesis (ko00940) pathway in each comparison group, indicating that phenylpropanoid biosynthesis (ko00940) was the most important pathway. Based on the above results, the genes and metabolites involved in this metabolic pathway were analyzed, and the results showed that the expression levels of most genes and the accumulation of metabolites changed significantly.

Monolignols are synthesized from phenylalanine through a series of enzymatic reactions catalyzed by phenylalanine ammonia lyase (PAL), cinnamic acid 4-hydroxylase (C4H), 4-coumarate: CoA ligase (4CL), cinnamoyl-CoA reductase (CCR), cinnamyl alcohol dehydrogenase (CAD), caffeoyl-CoA O-methyltransferase (CCoAOMT), ferulic acid 5-hydroxylase (F5H), and caffeic acid O-methyltransferase (COMT). In this study, we investigated the genes involved in phenylpropanoid biosynthesis pathway and the results showed that most of these genes showed low expression at 10 DAP, but high expression at 20 DAP and 30 DAP. In metabolomics analysis, phenylalanine, cinnamic acid, p-coumaric acid, p-coumaryl alcohol, caffeic acid, caffeyl aldehyde, coniferyl aldehyde, ferulic acid and sinapaldehyde were detected. There were significant differences in the accumulation of p-coumaric acid, p-coumaryl alcohol, caffeic acid and caffeyl aldehyde between Tif and Lps. Notably, the content of p-coumaryl alcohol (H-lignin) in Lps was significantly lower than that in Tif at 30 DAP and there was less accumulation of p-coumaric acid and caffeyl aldehyde in Lps at 20 DAP. Furthermore, the content of caffeic acid in Lps was significantly lower than that in Tif at 20 DAP and 30 DAP (Figure 6).

### 2.6. Genes Related to Monolignol Biosynthesis and Transport

Transcriptome data indicated that some genes encoding key enzymes of monolignol biosynthesis were differentially expressed. Metabolome data suggested that the accumulation of p-coumaryl alcohol (H-lignin) and caffeyl aldehyde were significantly different between Tif and Lps. Additionally, as precursors of coniferyl alcohol (G-lignin) and sinapyl alcohol (S-lignin), coniferyl aldehyde and sinapaldehyde were also detected, respectively. To further identify and clarify the crucial genes regulating the accumulation of these metabolites, we analyzed the correlation between the genes participating in monolignol biosynthesis and the content of p-coumaryl alcohol, caffeyl aldehyde, coniferyl aldehyde and sinapaldehyde, respectively (Figure 7A). The Pearson correlation analysis showed that the expression of *gene-LOC112771695* was highly correlated (0.90) with the amount of p-coumaryl alcohol, followed by *gene-LOC112717155*. *Gene-LOC112779615* and *gene-LOC112703807* were strongly correlated with the amount of coniferyl aldehyde, the rho values of the Pearson correlation coefficients were 0.98 and 0.92, respectively. The content of sinapaldehyde was highly correlated with the expression of *gene-LOC112696529* (−0.95) and *gene-LOC112759199* (−0.92). In addition, strong correlations were observed between the caffeyl aldehyde content and the expression of *gene-LOC112782630* (−0.87) and *gene-LOC112737203* (0.85).

To further understand monolignol fluxes, we studied the expression of the genes involved in their transport. ABC transporters participate in the transmembrane transport of monolignols [24]. We identified 136 genes encoding ABC transporters and analyzed their expression patterns. Meanwhile, canonical correlation analysis (CCA) of key metabolites (p-coumaryl alcohol, coniferyl aldehyde, sinapaldehyde and caffeyl aldehyde) and the expression of ABC transporter-related genes was performed (Figure 7B). The results showed that the expression of *gene-LOC112696487* was highly correlated with the amount of p-coumaryl alcohol. Therefore, it was a candidate gene for the transport of monolignols. Similarly, strong correlations were observed between the coniferyl aldehyde content and the expression of *gene-LOC112775360*, and sinapaldehyde content had a strong correlation with the expression of *gene-LOC112722707*. The expression of *gene-LOC112702885* was highly correlated with the amount of caffeyl aldehyde.

### 2.7. Validation of Candidate DEGs by qRT-PCR Analysis

Six and eight genes involved in monolignol biosynthesis and encoded ABC transporters were selected, respectively, and qRT-PCR was used to analyze their expression to verify the transcriptome data sets from RNA-Seq. The results revealed that the expression trend of these 14 genes quantified by the qRT-PCR were consistent with those obtained from RNA-sSeq analysis (Appendix A), thus verifying the reliability of RNA-Seq analyses.

### 2.8. Validation of the Candidate Metabolite in Different Peanut Varieties

To further validate the candidate metabolite linked to peanut pod size, we first investigated pod sizes of 89 different peanut varieties (Appendix A), then selected ten varieties with extreme pod sizes, and their pegs were marked in the same method as Tif and Lps. The ten varieties were divided into two groups (big pod and small pod). At 30DAP, the H-lignin content of these ten varieties and Tif (small pod) and Lps (big pod) was quantified. The results showed that the H-lignin content of peanut varieties with smaller pod sizes was significantly higher than that of peanut varieties with larger pod sizes at 30 DAP (Figure 8).

## 3. Discussion

Metabolites are tightly related to plant phenotypes [25], as compounds that accumulate in plant organs and affect plant phenotypes [26]. In peanut, there have been several comprehensive studies of the metabolomics responses to stresses such as salt [27], drought [28] and cold stress [29]. However, metabolomics studies on peanut pod development have been scarcely reported so far. During fruit development and ripening, metabolic processes result in changes in the fruit size [30]. Previous studies suggested that fruit peels contain high concentrations of flavonoids and phenolic acids [31]. In this study, the metabolomic analysis of peanut pods was performed for the first time, which considerably enriched the information concerning the metabolites of developing peanut pods. Our results showed that the major metabolites in peanut pods were lipids followed by phenolic acids and that flavonoids were the major DAMs between Tif and Lps at three growth stages. Moreover, the accumulation of metabolites in peanut pods was stage specific. During the 10 DAP to 20 DAP phase, the DAMs of peanut pods (Tif10DAP vs. Tif20DAP and Lps10DAP vs. Lps20DAP) were mainly phenolic acids, followed by lipids, organic acids, terpenoids, and amino acids and derivatives. From 20 DAP to 30 DAP, the main DAMs (Tif20DAP vs. Tif30DAP and Lps20DAP vs. Lps30DAP) were phenolic acids, followed by lipids, organic acids, nucleotides and derivatives, and flavonoids (Appendix A). These results showed that the accumulation of metabolites in peanut pods depended on developmental processes and that phenolic acids, lipids and flavonoids were the main metabolites affected pod development.

There are considerable changes in the transcriptional level during peanut pod enlargement [9]. In this study, DEGs between Tif and Lps pods were detected at 10, 20 and 30 DAP. Results showed that the number of DEGs showed a trend of rising first and then falling as development proceeded. GO enrichment analysis revealed that a considerable number of DEGs were correlated with cell wall organization or biogenesis, which was consistent with previous studies [21]. Xyloglucan endotransglucosylase/hydrolase (XTH) is a subgroup of the glycoside hydrolase family 16 (GH16) [32], which is involved in the modification of cell wall components and regulating plant growth and development [33]. Several studies demonstrated that XTH as a cell growth promoter is often correlated with cell expansion [34,35,36], as well as promoting fruit ripening [37]. In this study, most *XTHs* were upregulated in Lps at 10 and 30 DAP (Appendix A), and the pod sizes of Tif and Lps happened to be quite different in these two stages. This suggested that the differential expression of these *XTHs* might be associated with the difference in pod size. In addition, we also noticed that the phenylpropanoid biosynthetic process and metabolic process were major biological process categories of DEGs. Monolignols are biosynthesized from phenylalanine via the phenylpropanoid biosynthesis pathway and this process involves a series of enzymatic reactions. In this study, the differential expression of *PAL*, *C4H*, *4CL*, *CCR*, *CAD*, *COMT*, and *F5H* might result in differences in lignin content.

Just after cellulose, lignin is the second most abundant terrestrial biopolymer [38] and can be classified into three categories: para-hydroxy-phenyl lignin (H-type lignin), guaiacyl lignin (G-type lignin), and syringyl lignin (S-type lignin) [39]. It may not only provide mechanical support for plants [40], but also facilitate the conduction of water and minerals throughout the plant body [41], playing an important role in plant defense and growth [42]. Lignin is synthesized by the phenylpropanoid pathway, and the multi-omics data generated in this study demonstrated that the phenylpropanoid biosynthesis pathway was significantly enriched at all stages. At the transcriptional level, most genes associated with monolignol biosynthesis were upregulated in Tif at 30 DAP, which could mean that more monolignols accumulated in Tif. Metabolomics data confirmed this speculation that p-coumaryl alcohol, as the precursor of H-type lignin, accumulated considerably in Tif at 30 DAP. Meanwhile, we found that during the 10–20 DAP phase, the contents of p-coumaryl alcohol decreased in Tif and Lps, while the pod length and width of both Tif and Lps grew rapidly at this stage. Subsequently, the contents of p-coumaryl alcohol in Tif and Lps were increased significantly during the 20–30 DAP phase, and the growth-rate of their pod length and width slowed down. It is worth noting that less p-coumaryl alcohol was accumulated in Lps, and the slowing down of its growth trend of pod length and width was not as obvious as Tif (Appendix A). This indicated that the different accumulations of p-coumaryl alcohol might lead to different growth rates of Tif and Lps pods, resulting in different pod sizes. A previous study showed that tyrosine could be directly converted into p-coumarate by a grass bifunctional phenylalanine and tyrosine ammonia-lyase (PTAL) to produce monolignols [43]. However, in our study, there was no significant difference in the content of tyrosine and the expression of *PTAL* between Tif and Lps. These results suggest that p-coumaryl alcohol was the main monolignol that differed in abundance between Tif and Lps, and was a key metabolite affecting peanut pod size and the quantitative results of H-lignin contents at 30 DAP in peanut varieties with different pod sizes supported this view.

Lignin is synthesized by the oxidation polymerization of monolignols [44], and monolignols are produced by phenylpropanoid biosynthesis which is carried out within the plant cell [45]. However, the polymerization of the monolignols is performed in the apoplastic space, which requires the transmembrane transport of the monolignols [46]. Monolignols are moved by ABC transporters to the cell wall via the plasma membrane. The *gene-LOC112696487*, an ABC transporter-encoding gene, was found to be highly correlated with p-coumaryl alcohol, thus might be responsible for the transport of monolignols in this study. As a storage or transport form of monolignols [47,48], monolignol glucosides are thought to be sequestered in vacuoles [49]. In this study, the monolignol glucosides coniferin and syringin showed accumulation patterns the opposite of those of p-coumaryl alcohol at 30 DAP (Appendix A): less coniferin and syringin accumulated in Tif than Lps. Thus, we speculated that larger amounts of monolignol in the Tif would be transported to the cell wall and polymerized into lignin, while most of the monolignol in the Lps with a larger pod size would be glycosylated into monolignol glucosides and then stored.

## 4. Materials and Methods

### 4.1. Plant Materials

Two peanut varieties with contrasting pod size were used in this study: Tifrunner and Lps. The smaller pod size variety is Tifrunner (the reference genome of peanut, hereafter referred as Tif). Lps, with a larger pod size, is the core parent used for breeding high-yield peanut germplasm in Northern China, showing exciting potential for breeding purposes in the long-term breeding work. Both Tif and Lps lines were planted in the same areas (Laixi, Shandong, China). The peg that had not penetrated the soil of Tif and Lps were tied with colored tags, respectively. After marking, the soil was covered to ensure that the pegs were buried. Subterranean pods were collected from plants grown in the field at 10, 20 and 30 DAP. Pod samples were then rapidly frozen in liquid nitrogen and stored at −80 °C until metabolome and transcriptome analysis. Pod length and width were measured by Vernier caliper (five biological replicates for each material).

### 4.2. Widely Targeted Metabolomics Analysis

Sample preparation, extraction analysis, and qualitative and quantitative analysis of metabolites were carried out at Wuhan MetWare Biotechnology Co., Ltd., Wuhan, China. Biological samples were freeze-dried by vacuum freeze-dryer (Scientz-100F). The freeze-dried sample was crushed using a mixer mill (MM 400, Retsch) with a zirconia bead for 1.5 min at 30 Hz. Samples of 100 mg of lyophilized powder were dissolved with 1.2 mL 70% methanol solution, vortexed 30 s every 30 min, 6 times in total and placed in a refrigerator at 4 °C overnight. Following centrifugation at 12,000 rpm for 10 min, the extracts were filtrated (SCAA-104, 0.22 μm pore size; ANPEL, Shanghai, China, http://www.anpel.com.cn/, accessed on 1 November 2021) before UPLC-MS/MS analysis.

The sample extracts were analyzed using a UPLC-ESI-MS/MS system (UPLC, SHIMADZU Nexera X2, https://www.shimadzu.com.cn/; accessed on 5 November 2021; MS, Applied Biosystems 4500 QTRAP). The analytical conditions were as follows, UPLC: column, Agilent SB-C18 (1.8 μm, 2.1 mm × 100 mm); the mobile phase consisted of solvent A = pure water with 0.1% formic acid, and solvent B = acetonitrile with 0.1% formic acid. Sample measurements were performed with a gradient program that employed the starting conditions of 95% A, 5% B. Within 9 min, a linear gradient to 5% A, 95% B was programmed, and a composition of 5% A, 95% B was kept for 1 min. Subsequently, a composition of 95% A, 5.0% B was adjusted within 1.1 min and kept for 2.9 min. The flow velocity was set as 0.35 mL per minute; the column oven was set to 40 °C; the injection volume was 4 μL. The effluent was alternatively connected to an ESI-triple quadrupole-linear ion trap (QTRAP)-MS, with the following operating parameters: source temperature 550 °C; ion spray voltage (IS) 5500 V (positive ion mode)/−4500 V (negative ion mode); ion source gas I (GSI), gas II(GSII), curtain gas (CUR) were set at 50, 60, and 25.0 psi, respectively; the collision-activated dissociation(CAD) was high. Instrument tuning and mass calibration were performed with 10 and 100 μmol/L polypropylene glycol solutions in QQQ and LIT modes, respectively. QQQ scans were acquired as MRM experiments with collision gas (nitrogen) set to medium. DP and CE for individual MRM transitions was carried out with further DP and CE optimization. A specific set of MRM transitions were monitored for each period according to the metabolites eluted within this period. The data filtration, alignment and calculation were performed using Analyst 1.6.3 software (AB Sciex). Metabolite identification was based on the Metware MWDB database.

Unsupervised PCA (principal component analysis) was performed by statistics function prcomp within R (www.r-project.org, accessed on 2 March 2022). Cluster analysis of total metabolites was performed using the “pheatmap” package of R software. Significantly regulated metabolites between groups were determined by VIP ≥ 1 and absolute log2FC (fold change) ≥ 1. VIP values were extracted from the OPLS-DA result, which also contained score plots and permutation plots, generated using R package MetaboAnalystR. The data was log transform (log2) and mean centered before OPLS-DA. In order to avoid overfitting, a permutation test (200 permutations) was performed.

### 4.3. Transcriptome Sequencing

We selected 10 representative pods from Tif and Lps for each biological library construction (three biological replicates for each time point). RNA-Seq was performed using RNA extracted from peanut pods using RNAprep Pure Plant Plus Kit (TIANGEN BIOTECH, Beijing, China). The RNA concentration and RNA integrity were detected by Qubit RNA Assay Kit in a Qubit 2.0 Flurometer (Life Technologies, Carlsbad, CA, USA) and the RNA Nano 6000 Assay Kit of the Bioanalyzer 2100 system (Agilent Technologies, Santa Clara, CA, USA), respectively. Using fastp v 0.19.3 to filter the original data, mainly to remove reads with adapters, clean reads were mapped to the reference genome sequence (*Arachis hypogaea* cv. Tifrunner) (https://www.peanutbase.org/data/public/Arachis_hypogaea/, accessed on 15 October 2021) using HISAT v2.1.0. FeatureCounts v1.6.2 was used to calculate the gene alignment; then, the FPKM of each gene was calculated based on the gene length. DESeq2 v1.22.1 was used to analyze the differential expression between the two groups, and the *p*-value was corrected using the Benjamini–Hochberg method. The corrected *p*-value and |log2foldchange| were used as the threshold for significant differential expression. The enrichment analysis was performed based on the hypergeometric test. For KEGG, the hypergeometric distribution test was performed with the unit of pathway; for GO, it was performed based on the GO term.

### 4.4. Quantitative Real-Time PCR (qRT-PCR)

The samples used for qRT-PCR analysis were the same as RNA-Seq. Gene-specific primers for qRT-PCR are shown in Appendix A. The qRT-PCR was conducted using a SYBR Premix Ex Taq kit (TaKaRa, Dalian, China) following the manufacturer’s instructions. The amplification conditions were as follows: predenaturation at 95 °C for 10 min, denaturation at 95 °C for 15 s, and annealing and extension at 60 °C for 30 s. Fluorescence signals were collected during annealing and extension and the whole process was repeated for 40 cycles. To determine the relative expression of each gene among different samples, the 2^−ΔΔCt^ method was used along with the internal reference actin gene to normalize the results.

### 4.5. H-Lignin Quantification

Pod samples were pretreated according to previously reported methods [50,51]. The sample was dried and ground into powder. About 0.05 g of the sample was weighed and dissolved in 5 mL of reagent I (2 M NaOH) and 0.8 mL of reagent II (nitrobenzene) in a brown closed glass bottle, heated at 110 °C for 48 h, and the sample was cooled to room temperature. The sample was transferred to a 15 mL EP tube and extracted twice with reagent three (chloroform). The aqueous phase was acidified with reagent four (6M HCl) to a pH of about 2, and then extracted twice with reagent three (ethyl acetate) to merge the organic phase. The sample was dried by nitrogen blowing, and the methanol was diluted to 0.5 mL. The needle filter was filtered in a sample bottle with a lined tube and high-performance liquid chromatography was used for analysis. The mobile phase was a 0.1% solution of 30:70 (*v*/*v*) methanol/phosphoric acid in water, with a flow rate of 1.0 mL/min. Quantification of the H-lignin was carried out at 290 nm using the p-hydroxybenzaldehyde standards.

### 4.6. Statistical Analysis

Data were analyzed using Microsoft Excel and plotted using GraphPad Prism 8.0.2 and OriginPro 2021b software. Statistical analyses were performed using SPSS 17.0 software (SPSS, Inc., Chicago, IL, USA). The PCA plot, clustering heatmap, volcano plots and canonical correlation analysis (CCA) were made using the Metware Cloud Platform (https://cloud.metware.cn/#/tools/tool-list, accessed on 3 March 2022). The UpSet plot was made using TBtools software. Gene network diagram and correlation analysis were performed using the OmicStudio tools at https://www.omicstudio.cn/tool, accessed on 5 March 2022).

## 5. Conclusions

In this study, the metabolites of peanut pods were determined by widely targeted metabolomics, including 720 metabolites classified into 11 classes. The multi-omics analysis of the peanut pods revealed significant differences in the phenylpropanoid biosynthesis (ko00940) pathway among peanuts with contrasting pod sizes. Further analysis showed that p-coumaryl alcohol (H-lignin) was the candidate metabolite affecting pod sizes. Meanwhile, the content of H-lignin in genetically diverse peanut varieties further supported our findings. Furthermore, within DEGs related to monolignol biosynthesis and transport, the expression of *gene-LOC112771695* (4CL) and *gene-LOC112696487* (ABC transporter) were highly correlated with the amount of p-coumaryl alcohol, respectively, indicating that they may be the key factors involved in the regulation of pod size. Our study was the first attempt to investigate metabolomics of peanut pods and an informative multi-omics dataset was provided to facilitate the further identification of candidate genes involved in the determination of peanut pod size.

## Figures and Tables

**Figure 1 ijms-23-13533-f001:**
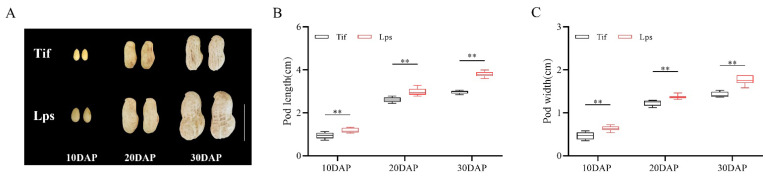
Phenotypes of Tif and Lps lines during pod expansion. (**A**) The phenotypic characteristics of Tif and Lps pods at three growth stages. Scale bar is 2 cm. (**B**) Pod length of Tif and Lps lines at three growth stages. (**C**) Pod width of Tif and Lps lines at three growth stages. ** *p* < 0.01.

**Figure 2 ijms-23-13533-f002:**
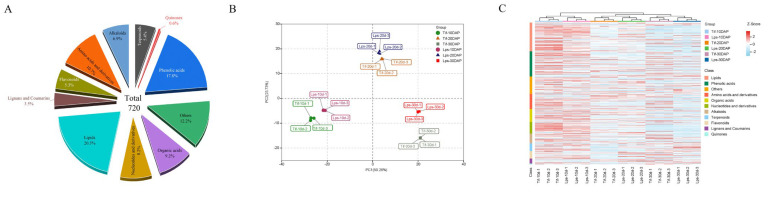
Metabolomic data analysis in peanut pod. (**A**) Functional classification of all detected metabolites. (**B**) Principal component analysis (PCA) of metabolites. (**C**) Clustering heatmap of the metabolites during pod development.

**Figure 3 ijms-23-13533-f003:**
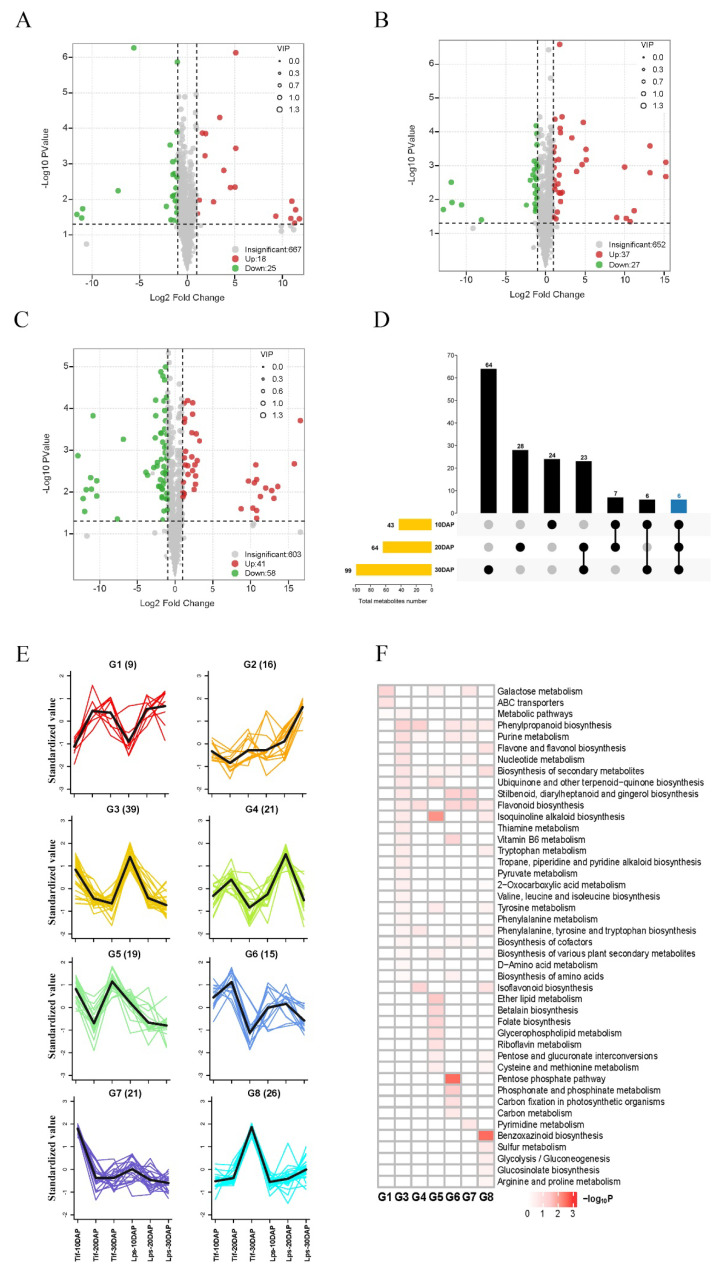
Volcano plots of accumulated metabolites between Tif and Lps (**A**) at 10 DAP; (**B**) at 20 DAP; (**C**) at 30 DAP. Variable importance in projection (VIP) ≥ 1 and |Log2FC| ≥ 1 were considered as DAMs. The larger the radius of the circle, the higher the value of VIP. (**D**) UpSet plot of three different comparison groups. The vertical axis represents the number of DAMs. (**E**) k-means clustering analysis on DAMs. Numbers in parentheses represent the number of DAMs per cluster. Each color line corresponds to a metabolite. The *x* axis represents Tif and Lps at three growth stages and the *y* axis depicts the standardized value of relative metabolite content. (**F**) KEGG enrichment analysis for DAMs of the 8 clusters.

**Figure 4 ijms-23-13533-f004:**
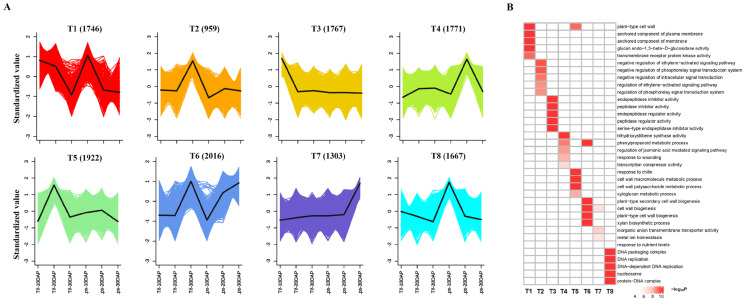
Global expression profile of three growth stages in Tif and Lps pods. (**A**) k-means cluster analysis on DEGs. Each color line corresponds to a gene. The x axis represents Tif and Lps at three growth stages and the y axis depicts the standardized value of RPKM per gene. (**B**) GO enrichment analysis of DEGs in the 8 clusters.

**Figure 5 ijms-23-13533-f005:**
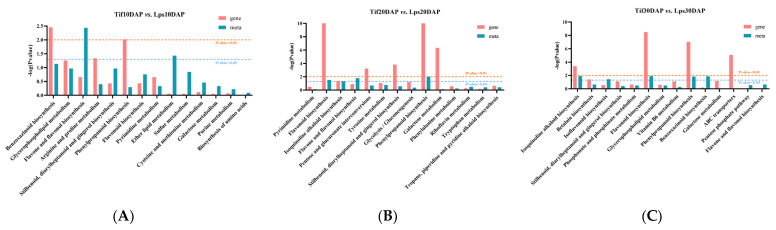
Joint KEGG enrichment *p*-value histogram of Tif vs. Lps at (**A**) 10 DAP, (**B**) 20 DAP and (**C**) 30 DAP.

**Figure 6 ijms-23-13533-f006:**
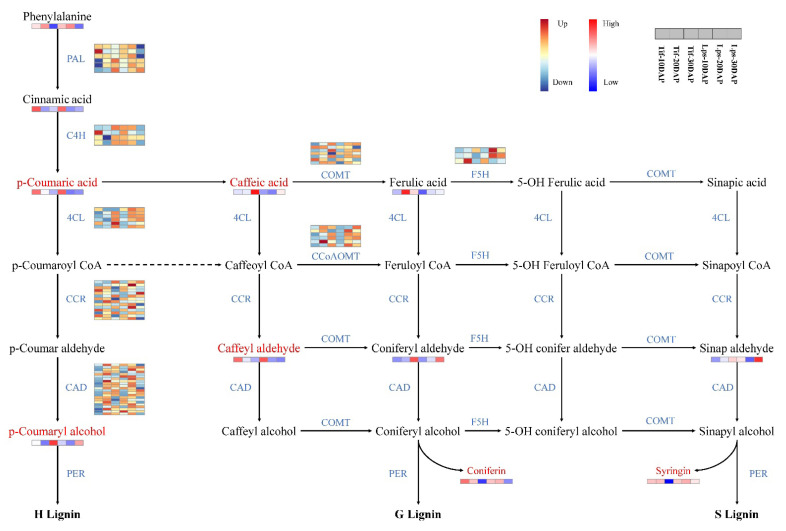
Transcriptomic and metabolic shifts in the phenylpropanoid biosynthesis pathway in Tif and Lps during the process of pod development. DAMs involved in the pathway are marked in red, and the relative content of each metabolite is displayed in the form of a heat map. The color scale indicates low (blue) to high (red) contents. For genes, heat maps represent the expression levels of DEGs in Tif and Lps; each row corresponds to a DEG, and the expression levels are shown in dark blue (downregulated) and dark red (upregulated).

**Figure 7 ijms-23-13533-f007:**
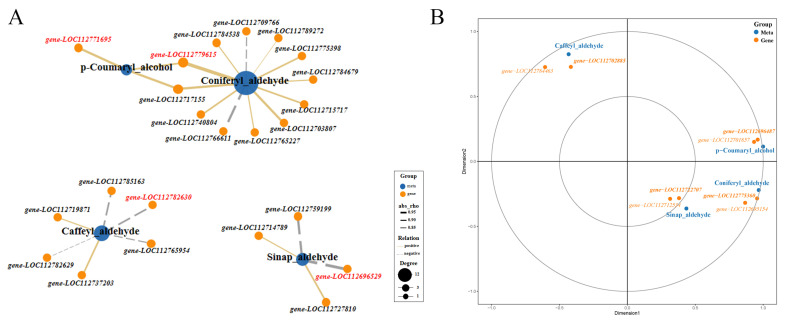
Pearson analysis of the correlation between metabolites and genes involved in monolignol biosynthesis and canonical correlation analysis (CCA) of metabolites and ABC transporter genes. (**A**) Correlation network graph was generated by considering only the strongest correlations (Pearson’s rho ≥ 0.80). Genes with the highest correlation with metabolites are highlighted in red. (**B**) CCA of metabolites and ABC transporter genes. In the figure, four regions are distinguished by the cross. In the same region, the farther away from the origin, the closer the mutual distance, and the higher the correlation. Metabolites are colored in blue, genes are colored in orange. Bold font indicates key genes.

**Figure 8 ijms-23-13533-f008:**
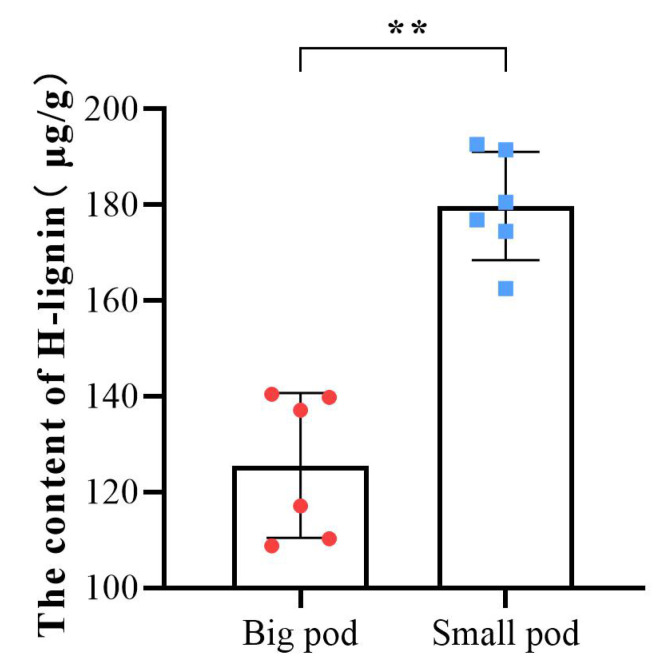
Determination of the H-lignin content in pods of 12 peanut varieties at 30 DAP. The color red corresponds to peanut varieties with larger pod sizes, and blue corresponds to peanut varieties with smaller pod sizes. Each point represents one variety. ** *p* < 0.01.

## Data Availability

The RNA-Seq data have been submitted to the NCBI Sequence Read Archive (SRA; http://www.ncbi.nlm.nih.gov/sra/, accessed on 21 April 2022) database with the accession number PRJNA828366.

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
