# Peer review of "Comparative Multi-Omics Analysis Reveals Lignin Accumulation Affects Peanut Pod Size"

_ijms, 2022, doi:10.3390/ijms232113533_

Round 1
Reviewer 1 Report
Congratulations for the figures and graphs made, they are very complex, but I suggest a better resolution of the images
I suggest a more detailed description of the plants material to be analyzed, after taking the samples, how were they stored until the analysis?
Author Response
First of all, thank you for your work and for helping me. Here is my reply to you:
Point 1: Congratulations for the figures and graphs made, they are very complex, but I suggest a better resolution of the images
Response 1: Thanks for your professional comments and suggestions. I have improved the resolution of lower resolution images.
Point 2: I suggest a more detailed description of the plants material to be analyzed, after taking the samples, how were they stored until the analysis?
Response 2: I have supplemented in the manuscript that the pod samples were quickly frozen in liquid nitrogen after sampling and stored at − 80 ° C until metabolome and transcriptome analysis.
Reviewer 2 Report
The work is very extensive and detailed with a lot of valuable information about metabolites composition in different peanut varieties with contrasting pod sizes in different developmental stages. Thus, major revision is necessary. First of all:
I would like to ask the authors to consider changing the title itself, so I suggest: “Comparative multi-omics analysis reveals lignin accumulation affects peanut pod size”.
Figures must be improved to be more understandable (Figures 3, 4, 5, 6).
Other suggestions and comments are given in the pdf file of the revised Manuscript.

Author Response
First of all, thank you for your work and for helping me. Here is my reply to you:
Point 1: I would like to ask the authors to consider changing the title itself, so I suggest: “Comparative multi-omics analysis reveals lignin accumulation affects peanut pod size”.
Response 1: Thanks for your professional comments and suggestions. I have revised the title according to your suggestion.
Point 2: Figures must be improved to be more understandable (Figures 3, 4, 5, 6).
Response 2: According to your suggestion, these figures have been modified accordingly.
Point 3: Other suggestions and comments are given in the pdf file of the revised Manuscript.
Response 3: I read the pdf file carefully and revised the manuscript according to the recommendations. I also uploaded a word document to answer your question.

Round 2
Reviewer 2 Report
The authors have revised the manuscript according to the given instructions, changed its title to a more suitable form, and minor text editing is in need for the manuscript to be published.